# Pharmacological Interventions for Pulmonary Involvement in Rheumatic Diseases

**DOI:** 10.3390/ph14030251

**Published:** 2021-03-10

**Authors:** Eun Ha Kang, Yeong Wook Song

**Affiliations:** 1Division of Rheumatology, Department of Internal Medicine, Seoul National University Bundang Hospital, Seongnam 13620, Korea; kangeh@snubh.org; 2Division of Rheumatology, Department of Internal Medicine, Seoul National University Hospital, Seoul 03080, Korea

**Keywords:** rheumatic, interstitial lung disease, pulmonary arterial hypertension, targeted therapy

## Abstract

Among the diverse forms of lung involvement, interstitial lung disease (ILD) and pulmonary arterial hypertension (PAH) are two important conditions in patients with rheumatic diseases that are associated with significant morbidity and mortality. The management of ILD and PAH is challenging because the current treatment often provides only limited patient survival benefits. Such challenges derive from their common pathogenic mechanisms, where not only the inflammatory processes of immune cells but also the fibrotic and proliferative processes of nonimmune cells play critical roles in disease progression, making immunosuppressive therapy less effective. Recently, updated treatment strategies adopting targeted agents have been introduced with promising results in clinical trials for ILD ad PAH. This review discusses the epidemiologic features of ILD and PAH among patients with rheumatic diseases (rheumatoid arthritis, myositis, and systemic sclerosis) and the state-of-the-art treatment options, focusing on targeted agents including biologics, antifibrotic agents, and vasodilatory drugs.

## 1. Introduction

Lung involvement is common in patients with rheumatic diseases (RDs), causing substantial morbidity and mortality in these patients. Individual RDs tend to be associated with a characteristic lung disease pattern where the key structure of the injury, as well as the critical cells and cytokines involved, are different [1].

Among the diverse forms of lung involvement, interstitial lung disease (ILD) and pulmonary arterial hypertension (PAH) are the two most important manifestations in patients with RD, leading to grave prognoses [2]. Of note, rheumatoid arthritis (RA), myositis, and systemic sclerosis (SSc) are the major systemic RDs, in which a significant proportion of the patients develop and die from ILD and/or PAH. However, recent advances in pharmacologic interventions have been shown to delay the disease progression of these lung conditions and also improve patient survival. In this review, we introduced updated knowledge on the treatment options for ILD and PAH in RDs, focusing on targeted therapies.

## 2. Interstitial Lung Disease (ILD)

Lung involvement in RDs most commonly takes the form of ILD. In particular, more than two thirds of the patients exhibit ILD in SSc and myositis [3,4]. Clinically symptomatic ILD is less frequent in RA than SSc and myositis, found only in about 10% of the patients [5]. However, the associated morbidity and mortality are never less. RD-associated ILD (RD-ILD) can be classified as usual interstitial pneumonia (UIP), nonspecific interstitial pneumonia (NSIP), organizing pneumonia (OP), diffuse alveolar damage (DAD), lymphoid interstitial pneumonia (LIP) and others, according to the radiologic and/or pathologic–morphologic patterns presented by the revised 2013 American Thoracic Society/European Respiratory Society classification of idiopathic interstitial pneumonias [6]. In RDs, the two most predominant types of ILDs are NISP and UIP [1]. While the prognosis is similar between RD-NISP and idiopathic NSIP, the prognosis of the RD-UIP other than RA-UIP is better than that of idiopathic UIP or idiopathic pulmonary fibrosis (IPF) [7].

### 2.1. Rheumatoid Arthritis Associated ILD (RA-ILD)

Potential targets of the lung injury in RA include almost all components of the lung structures. Thus, lung injury associated with RA encompasses a wide spectrum of disorders such as parenchymal (ILD), airway (bronchiectasis or bronchiolitis), pleural (pleurisy), and vascular diseases. Among them, ILD is most common.

#### 2.1.1. Clinical Features of RA-ILD

Approximately 10% of RA patients suffer clinically significant RA-ILD [5], with 8–9 times the lifetime risk of ILD development among RA patients compared to that of the general population (7.7% vs. 0.9%, respectively) [8]. Thirty-four percent of RA-ILD was found to occur within one year of RA diagnosis [9] and the risk of RA-ILD increases with RA duration and autoantibody (rheumatoid factor and/or anticitrullinated protein antibody) titers [10,11].

Unlike other RD-ILDs in which NSIP is the most prevalent histopathologic pattern, up to half of the RA-ILD cases show the UIP pattern, with NSIP as the second most common pattern [1]. With a heterogeneous progression rate across individuals, symptoms of RA-ILD usually progress over time once clinically present. The UIP patterns tend to demonstrate extensive disease at baseline and more rapid pulmonary function decline during follow-up, thus are associated with a worse prognosis [1,12]. The mortality of the patients with RA-ILD was as high as three times that of the patients with RA alone, with more than one third of RA-ILD patients being dead at five-years after ILD diagnosis [8,9].

Although it is considered that the natural progression of RA-ILD is heterogeneous, as in IPF, and a subgroup of those whose lung function progressively declines would show a grave prognosis, the exact natural history of RA-ILD is not fully known, particularly regarding acute exacerbation. In addition to the chronic lung function loss, acute exacerbation is another cause of the mortality associated with RA-ILD. However, a substantial proportion of patients with RA-ILD would suffer mixed patterns.

Acute exacerbation is a fatal condition characterized by a rapidly progressive respiratory failure. It has been well recognized not only in IPF and NSIP [6], but also in RD-ILD, particularly RA-ILD [13]. The radiographic characteristics of the acute exacerbation are defined as ground-glass opacities (GGOs) or consolidations newly overlaid on the background reticular abnormalities. Two thirds of RA-ILD patients with acute exacerbation died during the initial episode, causing a 2.5-fold increased mortality among RA-ILD patients [14]. The risk factors for the acute exacerbation of RA-ILD include UIP histology, old age, and methotrexate (MTX) use [14]. In IPF, more advanced lung fibrosis is also known as a reliable risk factor of the acute exacerbation [15].

#### 2.1.2. Pharmacologic Treatment of RA-ILD

Unfortunately, there have been no randomized controlled trials (RCTs) for RA-ILD and the EULAR and ACR recommendations do not specify how to treat RA-ILD yet [16,17]. In general, physicians either adopt the treatment strategies against the corresponding pattern of idiopathic interstitial pneumonia or practice empirical therapies with immunosuppressives. However, due to the heterogeneous clinical behaviors of RA-ILD, the treatment goal is hard to define: treatment to achieve recovery versus treatment to stabilize or slow progression. In the case of acute exacerbation, most therapies are immunosuppressants with potent anti-inflammatory effects, assuming reversibility of the lesions, while in case of chronic progression, the antifibrotic approach will take the priority [2]. Therefore, physicians need to understand the dominant pathogenic mechanism underlying the clinical behavior of individual cases of RA-ILD, which can vary at different time points even in the same patient.

##### Treatment with Conventional Agents

The data on the effect of individual immunosuppressive agents on RA-ILD are limited. Mycophenolate mofetil (MMF) was associated with a modest improvement in forced vital capacity (FVC) and the diffusing capacity of carbon monoxide (DLCO), reducing prednisone dosage in an observational study on 125 patients with RD-ILD including 18 RA-ILD cases [18]. In the case of the acute exacerbation of RA-ILD, high dose steroid therapy is delivered in combination with other immunosuppressive agents including MMF, azathioprine, or cyclophosphamide (CYC) [13,14].

##### Treatment with Targeted Agents

Although hope has been placed on biologics for RA-ILD treatment based on their excellent efficacy against articular inflammation, the evidence suggests that the safety profile of biologics including TNF inhibitors and others is uncertain for ILD treatment. Since biologics could, albeit rarely, exacerbate or cause the de novo development of ILD [19], experts recommend carefully assessing patients before and after treatment. While anti-TNF therapy has been suspected to aggravate ILD or induce its development [19], clear causal evidence is lacking [20,21]. Rituximab and abatacept have been more favorably suggested for patients with RA-ILD over anti-TNF therapy [22,23]. However, even these drugs have also been associated with lung toxicity and their data are scarce due to the less common use compared to TNF inhibitors [19].

Besides biologics, there are targeted agents with antifibrotic effects, recently introduced to treat IPF and RD-ILD (Table 1). These agents include nintedanib and pirfenidone.

Nintedanib is a competitive inhibitor of the nonreceptor and receptor tyrosine kinases that shows an antifibrotic effect [24]. The nonreceptor targets of nintedanib include Lck, Lyn, and Src and the receptor targets include platelet-derived growth factor receptors (PDGFR) α/β, fibroblast growth factor receptors (FGFR) 1/2/3, vascular endothelial growth factor receptors (VEGFR) 1/2/3, and fms-related tyrosine kinase 3 (FLT3). The targets of both categories play important roles in fibrosis [24]. In the phase II proof-of-concept TOMORROW study on patients with IPF diagnosed based on biopsy and/or high resolution computed tomography (HRCT) whose % predicted FVC of ≥50% and DLCO of 30–79% [25], 52-week nintedanib treatment (150 mg twice/day) was associated with less FVC decline, fewer acute exacerbations, and the preservation of health-related quality of life compared to placebo in patients with IPF. The antifibrotic effect of nintedanib on IPF was also replicated in phase 3 INPULSIS 1 and 2 trials, which showed significant reductions in the annual FVC decline in the treated group compared to the placebo (between-group difference of 125.3 and 93.7 mL/year in INPULSIS 1 and 2 trials, respectively, *p* < 0.001 in both results) [26]. Unlike treatment consistency on lung function preservation, however, the benefits on acute exacerbation were only observed in the INPULSIS 2. Nintedanib reduced the FVC decline among diverse subgroups defined by sex, age (<65 or ≥65 years), race (White or Asian), the baseline FVC% predicted (≤70% or >70%) and DLCO% (≤40% or >40%), composite physiologic index at baseline (≤45 or >45), the total baseline score on St George’s Respiratory Questionnaire (≤40 or >40), smoking status (never-smoker or current/ex-smoker), corticosteroids for systemic use at baseline (yes or no), and bronchodilator use at baseline (yes or no) [27,28].

The recent INBUILD study, an RCT that assessed the efficacy and safety of nintedanib in patients with non-IPF ILD (entry criteria: % predicted FVC of ≥45% and DLCO of 30–79%), included a subgroup of patients with RD-ILDs, mostly RA-ILD [29]. In the RD subgroup, nintedanib reduced the rate of FVC decline compared to placebo at 52 weeks with a between-group difference of 104 mL/year (*p* < 0.001) [29]. Based on the INBUILD trial, nintedanib was approved by the U.S. Food and Drug Administration (FDA) for progressive-ILD, including RA-ILD in March 2020.

Pirfenidone, another antifibrotic drug approved for IPF, might show similar benefits for patients with the RA-UIP as the drug did for patients with IPF. Unlike nintedanib, the mechanism of the action of pirfenidone is not well known. Among the three phase 3 clinical trials in patients with IPF [30,31], the Japanese trial showed that pirfenidone reduced the decline in vital capacity at week 52 (between-group difference, 0.07 L/year) and improved progression-free survival compared to placebo [30]. In the remaining two CAPACITY trials (004 and 006) on patients with multinational backgrounds (entry criteria: % predicted FVC of 50–90% and DLCO of 35–90%), the primary endpoint (% predicted FVC change from baseline to week 72) was met in trial 004 (−8.0% vs. −12.4%, *p* < 0.05) but not in 006 (−9.0% vs. −9.6%, *p* = 0.501) [31]. In the ASCEND trial [32], there was a relative reduction of 47.9% in the proportion of IPF patients who had an absolute decline of ≥10% in the FVC or who died (16.5% vs. 31.8%, *p* < 0.001). There was also a relative increase of 132.5% in the proportion of patients with no decline in FVC (22.7% vs. 9.7%, *p* < 0.001). In another multinational clinical trial on non-IPF ILD, the use of pirfenidone was associated with a lower risk of FVC decline greater than 10% (odds ratio = 0.44, 95% CI: 0.23–0.84) with a between-group difference in FVC of 95.3 mL/year at 24 weeks [33]. TRAIL1 is a 52-week multicenter randomized, double-blind, placebo-controlled, phase 2 study (ClinicalTrials.gov number, NCT02808871) of the safety, tolerability, and efficacy of pirfenidone in patients with RA-ILD, aiming to enroll as many as 270 subjects with RA-ILD and the results are awaited [34].

There is an overlap of adverse event profiles between nintedanib and pirfenidone [35]. In patients treated with nintedanib, gastrointestinal adverse events (e.g., diarrhea) were most common with mild-to-moderate intensity, accounting for the majority of the drug discontinuation [25,26,29]. Similarly, the most common adverse event of pirfenidone was gastrointestinal including nausea and vomiting, experienced by 40% and 18% of treated patients, respectively, out of 2059 person-years of exposure [36]. However, the two drugs have different pharmacokinetic profiles. Nintedanib is metabolized predominantly by ester cleavage and then glucuronidated to be excreted via the biliary system [35]. The use of nintedanib is associated with liver function test abnormalities in less than 5% of patients, and is not recommended for those with moderate-to-severe hepatic dysfunction. Regular liver function monitoring is required. Nintedanib has a low potential for drug–drug interactions, especially with drugs metabolized by cytochrome P450 enzymes. Pirfenidone is metabolized by various cytochrome P450 enzymes in the liver and predominantly excreted via the urine [35]. Similar rates of liver function test abnormalities were observed with pirfenidone as with nintedanib [25,26,31,32]. Pharmacokinetically, no drug–drug interaction was observed between nintedanib and pirfenidone. There are RCTs that examined the effect of nintedanib added on pirfenidone treatment [37,38]. More reports of nausea and vomiting were observed with nintedanib added on pirfenidone than used alone. However, most of them were mild-to-moderate as with the single drug treatment and the combination did not provide a new safety signal.

##### General Treatment Strategy for RA-ILD

Due to the heterogeneous progression patterns across individuals, it is hard to define an optimal treatment strategy in RA-ILD. Although the baseline extent of lung injury has been acknowledged as the most reliable risk factor of both progression and acute exacerbation, the cut-off extent to initiate treatment is not well defined. The international guidelines on RA management have yet to specify when to initiate treatment and what to be the first line agent [16,17]. Moreover, the treatment of acute exacerbation should be different from that of chronic progression [2]. However, it seems reasonable to use nintedanib when FVC loss is progressive enough to deteriorate symptoms (e.g., dyspnea or exercise capacity) or of when the current status is severe enough to qualify for the previous RCTs including the INBUILD trial (FVC of ≥45%, DLCO of 30–79%) [29].

### 2.2. Systemic Sclerosis Associated Interstitial Lung Disease (SSc-ILD)

The pulmonary manifestations of SSc can be both direct and indirect. The former category includes parenchymal (ILD) and vascular diseases (PAH) and the latter includes aspiration due to gastroesophageal reflux associated with esophageal sphincter fibrosis.

#### 2.2.1. Clinical Features of SSc-ILD

The key pathophysiology of SSc encompasses a triad of immune activation, vasculopathy, and fibrosis. Due to fibrosis and vasculopathy, the lung involvement of SSc most often manifests as ILD and/or PAH. ILD develops more frequently in the diffuse than limited cutaneous subset, particularly in the presence of antitopoisomerase I antibodies. SSc-specific anti-U3 RNP and anti-Th/To are also associated with SSc-ILD [39].

As many as 90% of SSc patients show ILD [3]. Among 3656 SSc patients from the European Scleroderma Trials and Research group, ILD was observed by plain chest radiography in approximately half of the patients with the diffuse cutaneous SSc and in one third of the patients with the limited cutaneous SSc [40]. Regarding severity, moderate (FVC of 50–75%) to severe (FVC < 50%) restrictive lung disease was found in 40% of SSc patients [41]. The pulmonary function of SSc-ILD patients mostly declines during the first several years after the onset of non-Raynaud’s symptoms, but the progression rate of decline is best predicted by the baseline severity of the lung function or fibrosis [1]. A greater impairment at baseline predicts more progression in the future. Further, there are a significant proportion of patients not treated without progression [42], suggesting that the clinical course of SSc-ILD is variable [43].

The predominant histologic pattern of SSc-ILD is NSIP based on biopsy and/or HRCT [1,3,44]. However, the histologic pattern does not affect the clinical outcome of SSc-ILD [1,7,44]. Because fibrosis is the main histologic finding other than inflammation, even GGO represents fine reticulation and is rarely reversed but replaced later by overt fibrotic findings such as reticulation or honeycombing [45,46].

The mortality of SSc patients showed an overall three-fold increased standardized mortality ratio compared to the general population [47]: the survival of SSc patients was 74.9% at 5 years and 62.5% at 10 years from the diagnosis, with the mortality risk in the presence of ILD being 2.9-fold compared to in the absence of ILD [47].

#### 2.2.2. Pharmacologic Treatment of SSc-ILD

The lung function of SSc patients with FVC of ≥80% at baseline rarely declines [43]. Thus, symptomatic patients are the primary target of treatments, particularly those whose ILD is moderate-to-severe or shows progression. The recent European consensus statements presented an agreement on screening of ILD for all SSc patients at baseline particularly in the presence of risk factors (diffuse cutaneous subset, antitopoisomerase antibody, low DLCO) preferably by HRCT but also with pulmonary function test (PFT) and clinical assessment as supporting tools [48]. The statements recommend treatment for all severe cases defined by PFT or HRCT, and for those who progress based on clinical assessment, HRCT, and/or PFT. However, the statements did not specify the threshold to define severe ILD or when to initiate or escalate treatment. Those who have symptoms that are progressively deteriorating (newly developed functional class II and more), whose FVC and/or DLCO decline is large enough (≥10% and ≥15%, respectively) [49,50], or whose FVC or DLCO is severe enough to justify treatment-related adverse events (refer to the inclusion criteria for Scleroderma Lung Study or SENSCIS trial below) [51,52,53] would be reasonable candidates to initiate or escalate treatment.

##### Treatment with Conventional Agents

Until recently, the treatment SSc-ILD has relied on immunosuppressants such as CYC and MMF based on the RCT data [51,52,54]. Since GGO of SSc-ILD often indicates fine fibrosis rather than inflammation [45,46], these immunosuppressive treatments stabilize rather than improve such lesions.

In the Scleroderma Lung Study (SLS) I on 158 SSc patients with symptomatic ILD (active alveolitis on imaging and FVC ranging between 45–85%) [51], the mean absolute difference of FVC% predicted at 12 months (2.53%, *p* < 0.03) was significantly in favor of oral CYC over placebo. When the patients were followed for another one year after stopping the medication, the effect was maintained or even greater at six months after treatment termination but disappeared by one year [52]. According to the subgroup analysis, a greater response to CYC was linked to more severe lung disease at baseline [52]. In SLS II, the efficacy of the two-year treatment with MMF was comparable to that of the one-year treatment with oral CYC at 24 months (mean FVC improvement from baseline of 2.17% in the MMF group versus 2.86% in the CYC group) [54]. The adverse events were less with the former drug. The 2017 updated EULAR recommendations still suggest CYC preferentially over MMF [55], but MMF is a viable first line treatment for patients who are susceptible to the toxicity of CYC.

##### Treatment with Targeted Agents


**Antifibrotics**


The SENSCIS trial was performed on 576 patients with SSc-ILD (mean age 54.6 years, 74–76% female) of at least 10% extent of the lung [53]. The entry criteria of ILD required % predicted FVC ≥ 40% and % predicted DLCO ranging in 30–90%. The % predicted FVC of included patients was 72% and DLCO 53% at baseline. Of the patients included in the study, 48.4% were taking background MMF. After 52 weeks of treatment, nintedanib (150mg twice daily) was associated with a lower annual rate of FVC decline compared to the placebo treatment (−52.4 mL/year versus −93.3 mL/year). The curves for the FVC change from the baseline separated by 12 weeks and continued to diverge until the end of study. The effect of nintedanib was additive when combined with MMF: the annual rates of change in FVC among the patients receiving mycophenolate were −40.2 mL/year in the nintedanib group and −66.5 mL/year in the placebo group, and the corresponding rates among the patients who were not receiving mycophenolate were −63.9 mL/year and −119.3 mL/year. Of note, the SENSCIS trial did not show any treatment effect on the skin fibrosis. According to the subgroup analysis, the effect of nintedanib was consistent regardless of the use of MMF, suggesting that there is no synergy but additive effect between nintedanib and MMF [56]. The gastrointestinal adverse (e.g., diarrhea) events were more common in the SENSCIS trial compared to the INPULSIS trials probably due to the gastrointestinal involvement by SSc itself [26,53]. The post hoc analysis on the SENSCIS trial showed that the effect of nintedanib was significant to prevent % predicted FVC decline of >10% but not of >5% [57]. An open label extension study is ongoing to provide long term data on nintedanib use in patients with SSc-ILD (ClinicalTrials.gov number, NCT03313180). Based on the SENSCIS trial [53], nintedanib was approved to treat SSc-ILD by the FDA in 2019 and by the European Medicines Agency (EMA) in 2020.

The acceptable safety and tolerability of the antifibrotic agent pirfenidone were reported among patients with SSc-ILD [58], but its efficacy needs to be assessed in further clinical trials. There are two ongoing RCTs on SSc-ILD. The efficacy and safety of pirfenidone is now being assessed in a phase 3 study on 144 SSc-ILD patients (ClinicalTrials.gov number, NCT03856853). In addition, the SLS III study (phase 2) will randomize 150 patients with SSc-ILD to pirfenidone versus placebo, with background MMF, over 18 months of follow-up (ClinicalTrials.gov number, NCT03221257), to assess the effect of combination treatment of pirfenidone and MMF versus MMF alone.


**Biologics**


Promising benefits in SSc-ILD from biologics have recently been suggested. In the phase II faSScinate trial that assessed the skin fibrosis as the primary outcome and lung function as the secondary outcome, 48-week treatment with tocilizumab (TCZ) provided an encouraging numerical improvement in the modified Rodnan skin scores and evidence of less decline in lung function [59]. During another 48-week open-label extension period of TCZ treatment, skin score improvement and FVC stabilization were observed in the placebo-treated patients who transitioned to tocilizumab [60]. In the faSScinate study, the patient population showing benefits from TCZ had a shorter duration of disease (mean disease duration, 1.6 years), increased serum acute phase reactants, progressive skin disease, and low normal mean FVC levels (mean predicted FVC%, 81%) at the study baseline. These findings may suggest that more inflammatory process is involved during the early phase of the disease, where immunosuppressive or anti-inflammatory treatment can provide more benefits. In the subsequent focuSSced study of the multinational background on 210 patients with early diffuse cutaneous SSc (duration ≤ 5 years) [61], TCZ and placebo were compared regarding skin fibrosis (primary outcome) and lung function (secondary outcome). After 48 weeks, the skin fibrosis endpoint was not met but the FVC loss was more in favor of the TCZ group than the placebo (between-group difference, 241 mL, *p* = 0.002). The results of the focuSSced study on FVC should be interpreted with caution because the SSc-ILD was not the primary outcome and only 65% of enrolled patients had SSc-ILD at baseline. Controlled studies are awaited to assess whether TCZ has an effect in patients with severe established SSc lung disease. In addition, the paradoxical response of aggravating ILD has been also reported from TCZ treatment [19], indicating that meticulous monitoring of ILD is mandatory before and after treatment despite the overall benefits of TCZ in patients with ILD.

Rituximab has also been used to treat SSc-ILD. In an open-label RCT on 60 patients with SSc-ILD, the RTX treatment showed a significantly better patient response than the CYC treatment in terms of improvement in FVC and skin fibrosis [62]. However, a propensity score matched observational study showed that the proportion of patients with FVC decline of ≥10% was similar regardless of RTX treatment. Nevertheless, patients treated with RTX were more likely to stop or decrease steroid use [63]. According to the experts, rituximab was considered as one of the first induction agents for SSc-ILD in addition to MMF or CYC [64]. An ongoing trial (RECITAL) is comparing rituximab and CYC for RD-ILD including SSc-ILD [65].

##### General Treatment Strategy for SSc-ILD

For those who are asymptomatic with minimal extent of ILD (e.g., FVC > 80%), watchful monitoring without treatment is justified [42,43]. For those who need treatment, the EULAR guidelines endorse CYC over MMF as the first line treatment for SSc-ILD [55] while the European consensus statements consider CYC and MMF, and nintedanib as equivalent options for treatment initiation or escalation [48]. The subgroup analysis of SENSCIS showed a comparable FVC change between placebo with baseline MMF group (−66.5 mL/year) versus nintedanib without baseline MMF group (−63.5 mL/year) [53]. Thus, whether to use antifibrotics as a first line treatment needs further data (e.g., head-to-head comparison between immunosuppressants versus antifibrotics) and discussion. A reasonable approach would be to switch to or add nintedanib when refractory to CYC or MMF. Although some evidence emerged suggestive of the benefits with TCZ and RTX on lung function, such evidence is only exploratory and needs to be confirmed in an RCT that examines lung function as the primary end point.

### 2.3. Myositis Associated Interstitial Lung Disease (Myositis-ILD)

As in SSc, the pulmonary manifestations of myositis can be categorized as direct and indirect. The direct involvements are mostly ILD. The indirect involvements consist of aspiration pneumonia due to pharyngeal muscle weakness and hypoventilation due to respiratory muscle weakness [66].

#### 2.3.1. Clinical Course of Myositis-ILD

A majority of myositis-associated lung involvement manifests as ILD. Unlike in SSc, isolated PAH is rare in myositis [67]. ILD manifests early in the course of myositis, often found at the time of diagnosis [4,68,69]. Being prevalent in up to two thirds of myositis patients [4,69], more than 90% of myositis patients develop ILD in the presence of antiaminoacyl tRNA synthetase (ARS) antibody [70]. The clinical course of myositis-ILD is diverse but distinguished from SSc-ILD in that as many as 20% of myositis-ILD cases manifest as a rapidly progressive form that challenges successful treatment [69]. This contrasts with SSc-ILD, where fibrosis is the hallmark of the disease, causing more chronic progression. Such difference indicates that inflammatory process is a dominant process in myositis-ILD, generally in proportion to symptom deterioration rate, and culminates in the rapidly progressive form in which respiratory failure can occur even within days [71,72].

While NSIP is most common for myositis-ILD in general [1], DAD is more common in case of the rapidly progressive type [70]. The rapidly progressive ILD tends to cluster in patients with either classic DM or amyopathic dermatomyositis than polymyositis, particularly in the presence of antimelanoma differentiation-associated gene 5 (anti-MDA5) [73]. Myositis-ILD of the rapidly progressive type is refractory to treatment leading to a high mortality [72].

The treatment response of myositis-ILD follows the underlying histological pattern [69,74,75]. OP is well treated by steroids alone, whereas DAD and UIP respond poorly even to combination therapy [69,74]. The treatment response of NSIP depends upon the degree of inflammation compared to fibrosis [75]. Biomarker studies showed that high serum levels of ferritin, IL-18, or soluble CD206 (macrophage mannose receptor) were associated with a poor treatment response among patients with anti-MDA5-positive ILD [76,77].

#### 2.3.2. Pharmacologic Treatment of Myositis-ILD

Due to the rarity of the disease and the heterogeneous clinical subsets within the disease, the treatment of myositis-ILD most often relies upon empirical therapy consisting of steroids combined with conventional immunosuppressives such as CYC, cyclosporine, or MMF without sufficient data supported by RCTs [1]. Even more limited are the data for targeted treatments. A recent Japanese study showed that the initial combination treatment with high dose steroid, tacrolimus, and CYC with or without plasmapheresis was associated with a better response in anti-MDA5-positive ILD patients compared to step-up therapy from initial high-dose steroids [78]. This finding also put emphasis on the early aggressive treatment for anti-MDA5-positive, rapidly progressive ILD. Moreover, a Spanish expert group agreed that anti-MDA5-positive ILD should be similarly treated [79]. Of note, despite the disease-specific anti-MDA5 autoantibody, successful treatment with rituximab has only been anecdotally reported [80].

### 2.4. Future Treatment Strategies for Rheumatic Disease Associated Interstitial Lung Disease

Nintedanib and pirfenidone slow but do not halt the progression of IPF. Thus, interest in the combination of the two drugs is growing, but more data are needed on the safety and efficacy of combination therapy. There are a few studies that showed relatively acceptable tolerability and safety of the two drugs combined for a short-term period (12~24 weeks) [37,38]: the combination was completed in two thirds of patients despite a higher adverse event rate compared to the single drug treatment. However, there have yet to be large randomized controlled trials assessing whether combination therapy has increased efficacy compared to treatment with a single antifibrotic drug.

Several clinical trials evaluating the efficacy of the new drugs on pulmonary outcomes are underway. The ISABELA 1 and 2 trials are now ongoing to examine the effect of autotaxin inhibitor GB-0998 among patients with IPF (ClinicalTrials.gov number, NCT03711162; NCT03733444) [81]. A phase 2 RCT of bortezomib plus MMF versus MMF alone on patients with SSc-ILD is currently ongoing (ClinicalTrials.gov number, NCT02370693). Romilkimab, a monoclonal antibody targeting IL-4 and IL-13, showed efficacy after 24 weeks on skin fibrosis associated with diffuse SSc under background immunosuppressive agents in a phase 2 pilot study [82]. However, it failed to show benefits in the of IPF after 52 weeks [83]. Guselkumab (monoclonal antibody that blocks IL-23) is under investigation in SSc patients with skin fibrosis as the primary outcome of and lung function as one of the secondary outcomes (ClinicalTrials.gov number, NCT04683029).

## 3. Pulmonary Arterial Hypertension (PAH)

Abnormal proliferation, vasoconstriction, and thrombosis of the pulmonary vasculature are the main pathogenic mechanisms of PAH. Right heart catheterization (RHC) is the gold standard to diagnose PAH using the criteria of a mean pulmonary arterial pressure of ≥25 mmHg and a pulmonary capillary wedge pressure of ≤15 mmHg [84].

### 3.1. Systemic Sclerosis Associated Pulmonary Arterial Hypertension (SSc-PAH)

Not only isolated PAH but also PH secondary to either left heart dysfunction or ILD progression can occur in SSc, either separately or in combination. The SSc-PAH shows a unique phenotype of PAH with a worst prognosis compared to idiopathic or non-SSc RD- PAH [85,86].

#### 3.1.1. Clinical Features of SSc-PAH

The mortality risk of SSc patients increases by 3-fold in the presence of PAH, showing PAH is a deadly complication of SSc patients [87]. Therefore, patients should be screened for PAH based on symptoms, echocardiography, and DLCO patterns and undergo RHC if identified as high-risk [88,89]. With RHC, SSc-PAH was found in approximately 10% of these high-risk candidates. The limited rather than the diffuse cutaneous subset shows a higher prevalence of SSc-PAH, which was observed in up to half of the patients with CREST syndrome [90]. It was reported that 50% of SSc-PAH occurs within five years from the first non-Raynaud phenomenon symptom with the mean interval from SSc diagnosis to PAH occurrence of 6.3 years [91].

Like those with idiopathic PAH, patients with SSc-PAH are either clinically silent or show only nonspecific symptoms until their disease is advanced. Therefore, active surveillance is the key to early intervention, which indeed led to better survival compared to passive identification [92]. In SSc patients, male sex, old age, overt vasculopathy such as telangiectasia and digital ulcers, anticentromere or anti-U3 RNP antibodies, and the limited cutaneous subset (e.g., CREST syndrome) were identified as risk factors for PAH [93]. However, both the sensitivity and the specificity of these risk factors are limited. Other findings suggestive of PAH include elevated levels of the N-terminal probrain natriuretic peptide (NT-proBNP) or disproportionately low DLCO [94,95]. In particular, echocardiographic measures are one of the important screening tools to identify candidates for RHC. For example, a tricuspid regurgitation (TR) velocity of ≥2.5 m/s is considered one of the findings highly suggestive of PAH [96]. However, the sensitivity at this TR velocity threshold is limited, missing 20% of the mild PAH patients [96,97]. DETECT is a multidimensional algorithm to identify SSc patients at a high risk of PAH, used as the active surveillance strategy to avoid the delayed diagnosis of PAH [1,96]. Among 57 DETECT enrollees with SSc-PAH, 44% progressed during a median follow-up of 12.6 months [98]. The factors associated with progression were male gender, disproportionately low DLCO, and poor functional capacity. A more simplified and practical algorithm proposed in 2013 to initially screen SSc patients uses PFT, echocardiography, and NT-proBNP for referral for RHC [99] (Table 2).

In a recent prospective observational study, 93 SSc-ILD patients were screened and underwent RHC following the above 2013 algorithm. Of these, 31.2% were found to have RHC-proven PAH, and the survival rate was 91% at three years [100]. This improved survival is striking compared to the exceptionally grave prognosis (survival of 39% at three years) previously reported for SSc patients with coexisting ILD and PAH [101]. Based on this finding, the poor prognosis of SSc-PAH patients compared to patients with idiopathic or PAH of other rheumatic diseases could be due to delayed diagnosis and treatment [85,86], and early treatment is paramount for better survival.

#### 3.1.2. Pharmacological Treatment of SSc-PAH

##### Treatment with Conventional Agents

Immunosuppression against the hyperimmune state of SSc is never sufficient to stop SSc-PAH progression due to fibrotic and proliferative pathways driven by nonimmune cells. An acute vasoreactive response is observed in 10% of the patients with RD-PAH, but the response to calcium channel blockers beyond 3–4 months is preserved in less than 1% of these patients [102].

##### Treatment with Targeted Agents and Risk Stratification

Although the autoimmune process may be the fundamental mechanism orchestrating the initiation and progression of RD-PAH, its downstream effects involve nonimmune components as well, leading to pulmonary vascular remodeling. In particular, the treatment effect of anti-inflammatory agents is limited in lung diseases associated with SSc, indicating that noninflammatory pathways play a dominant role. In PAH, endothelial dysfunction due to abnormally regulated vasoactive and/or proliferative mediators induces vascular constriction and remodeling. Currently, three pathways important in endothelial function are targeted by PAH treatments, endothelin-1, nitric oxide (NO), and prostacyclin pathways [103]. Endothelin receptor antagonist (ERA), phosphodiesterase 5 inhibitor (PDE5i), and soluble guanylate cyclase (sGC) stimulator, which potentiate the effect of nitric oxide (NO), as well as prostacyclin analogs (PCA), are commercially available (Table 3).

According to the 2017 EULAR recommendations [55], ERA (ambrisentan, bosentan, and macitentan), PDE5i (sildenafil and tadalafil), and an sGC stimulator (riociguats) are considered the first-line options for treating SSc-PAH based on high-quality RCTs that showed improvement in exercise capacity, the time to clinical worsening (defined as a composite of death, hospitalization, and disease progression), and/or PAH-related hemodynamics in heterogeneous patients with PAH including RD-PAH. These studies were not powered to show the independent efficacy of the SSc-PAH subgroup, but the EULAR recommendations are based on the extrapolation of the high-quality RCT results. In the subgroup analyses, the direction of the results was not different between patients with idiopathic PAH and RD-PAH. In the idiopathic PAH, significant improvements in exercise capacity and reductions in mortality (pooled effect, 44% reduction; *p* < 0.041) by these treatments were demonstrated [104].

A high-quality RCT showed that continuous intravenous epoprostenol (a prostacyclin analog) administration improved exercise capacity, functional class, and hemodynamics in patients with SSc-PAH [105]. The administration of continuous intravenous epoprostenol requires an indwelling central venous catheter and abrupt cessation of the drug may cause PAH rebound that can be fatal. Thus, the EULAR recommendations suggest that this treatment should be considered for severe SSc-PAH of class III and IV [55]. In addition to epoprostenol, other PCAs (iloprost and treprostinil) showed similar results in high-quality RCTs on heterogeneous patients with PAH including RD-PAH and are also approved for the treatment of PAH including RD-PAH [55]. Although not commented in EULAR recommendations, selexipag, an oral selective prostacyclin IP receptor agonist, has shown efficacy in the phase 3 GRIPHON trial of reducing the risk of a primary composite endpoint of morbidity and mortality by 40% and 41% in PAH and RD-PAH (majority, SSc-PAH), respectively [106,107]. Selexipag is the only oral prostacyclin receptor agonist that showed a reduction of morbidity and mortality in a RCT.

In addition to the application of individual drugs that target each corresponding pathway, the advancements in recent treatment strategies include the risk assessment of PAH patients and linking the baseline severity of PAH to the subsequent treatment intensity or escalation [97,103]. According to the 2015 European Society of Cardiology (ESC)/European Respiratory Society (ERS) pulmonary hypertension guidelines [97], a multiparametric approach should be considered to stratify patients into low-, intermediate- or high-risk groups for 1-year mortality using clinical (clinical signs of right heart failure, the progression of symptoms, and syncope), functional class (WHO or NYHA class), exercise (6-min walking distance and cardiopulmonary exercise testing), biochemical (NT-proBNP), and echocardiographic/hemodynamic parameters (Table 4).

Another important feature of the 2015 ESC/ERS guidelines for PAH is the endorsement of pre-emptive combination therapy targeting different endothelial pathways, particularly for high-risk patients [97]. The most compelling evidence for such strategy came from the recent AMBITION trial where a 50% reduction in the composite endpoint of clinical failure events at 24 weeks was seen from treatment with the upfront combination of ambrisentan and tadalafil compared to either drug alone in patients with PAH and RD-PAH [108]. Similarly, the add-on effect of a study drug (either macitentan, selexipag, or riociguat) was observed in other RCTs when combined with a background PAH treatment using a different class [106,109,110], also endorsing sequential combination therapy. Similar evidence for upfront or initial combination or sequential combination therapy was also observed in patients with RD-PAH including SSc-PAH [107,111,112]. The 2015 ESC/ERS guidelines also depict a treatment algorithm that recommends treatment escalation (double to triple, maximal medical therapy including intravenous PCA) when the reassessment in 3–6 months shows an insufficient response such as residual intermediate- or high-risk status [97]. However, the evidence for the benefits of combination therapy remains to be replicated, specifically for SSc-PAH in future studies.

## 4. Pulmonary Manifestations of Other Rheumatic Diseases

Other than RA, SSc, and myositis, Sjogren’s syndrome (SjS) and mixed connective tissue disease (MCTD) also show a substantial prevalence of pulmonary manifestations including ILD and PAH. Although the data on the epidemiology are growing in SjS and MCTD, RCT data regarding the treatments are very limited. However, we may be able to apply similar treatment strategies in SjS and MCTD as in other RDs described in this review, based on the dominant underlying mechanism of fibrosis versus inflammation or chronic progression versus acute exacerbation.

### 4.1. Sjogren’s Syndrome Associated Pulmonary Involvements

SjS is a systemic autoimmune disease primarily affecting exocrine glands, which ultimately leads to the destruction of the given tissue [113]. The autoimmunity of SjS activates both immune cells and glandular epithelial cells showing a histological lesion of focal lymphocytic infiltrates, enriched in CD4+ T-cells, around the salivary and lachrymal ducts and even in lung tissues [114]. Of note, the focus score of salivary glands has been shown to correlate with the increased prevalence of airway disease and ILD in SjS [115]. These findings may suggest that the glandular and extraglandular lesions of SjS share similar pathogenic pathways involving autoimmunity to epithelial cells.

The prevalence of pulmonary involvement in patients with SjS has been estimated to range from 10% to 20% [116]. Airway disease and ILD are the predominant forms of lung involvement among the pulmonary manifestations of SjS, but lymphoproliferative disorders as well as cystic lesions are also observed [116]. The most common histologic type of ILD associated with SjS was found to be NSIP followed by UIP and LIP [117,118], with NSIP present in up to 45% of biopsied cases [117] and mostly of the fibrotic type [118]. Unlike RA-UIP, SjS-UIP tends to have a better response to immunotherapy as compared to IPF [119]. LIP constitutes 15% of SjS-ILD, of which clinical course is variable from complete resolution without treatment to progression and possible death or transformation to lymphoma [120].

### 4.2. Mixed Connective Tissue Disease Associated Pulmonary Involvements

MCTD is characterized by mixed features of two or more RDs including but not limited to SSc, myositis or systemic lupus erythematosus, with disease specific high titer anti-U1 RNP antibodies [121]. The common pulmonary manifestations of MCTD are ILD and PAH. A Norwegian nationwide cross-sectional study showed that 35% of MCTD patients had lung fibrosis in HRCT after a mean disease duration 9 years [122]. A higher rate of ILD was found (up to two thirds of the patients) in the hospital-based study [123]. In the Norwegian study, 19% of MCTD associated ILD (MCTD-ILD) was severe in extent involving >50% of lung parenchyma [122]. However, the retrospective cohort study on unselected 53 patients with a mean disease duration of 9 years in Brazil showed that 51% of the patients had ILD at baseline with a mean FVC that remained stable at around 77% over 10 years of follow-up and a mean DLCO that declined from 84% to 71% [124]. Such discrepant findings leave a debate on the natural course of MCTD-ILD, and further cohort studies at a larger scale are needed. Together with ILD, PAH is a major prognostic factor of MCTD [125,126]. The Norwegian population-based prevalence of pulmonary hypertension was reported to be less than 5% over 5.6 years [125], but PAH has been recognized as the leading cause of death in patients with MCTD, explaining 41% of all deaths [126].

## 5. Conclusions

ILD and PAH are the two most important prognostic factors in patients with RD and are associated with significant morbidity and mortality. Early diagnosis and early treatment are the key steps to the successful management of these two conditions, improving survival. In addition to conventional immunosuppressants, new targeted treatments are under investigation and some have already been incorporated into the international recommendations for treating RD-related ILD or PAH. However, more data are needed to ensure that the efficacy and safety of drugs used to treat RD as a whole are demonstrated for a specific RD.

## Figures and Tables

**Table 1 pharmaceuticals-14-00251-t001:** Recent clinical trials on targeted drugs for idiopathic pulmonary fibrosis and rheumatic disease-associated interstitial lung disease.

Targeted Therapy	Mechanism of Therapy	Target Disease	Key RCT Name	Primary Outcome	Treatment Duration	Proven Efficacy
Nintedanib	Tyrosine kinase inhibition	IPF	TOMORROW	Annual FVC decline	52 weeks	Reduced FVC decline
		IPF	INPULSIS 1 and 2	Annual FVC decline	52 weeks	Reduced FVC decline
		Non-IPF ILD (e.g., RA-ILD)	INBUILD	Annual FVC decline	52 weeks	Reduced FVC decline
		SSc-ILD	SENSIS	Annual FVC decline	52 weeks	Reduced FVC decline
Pirfenidone	Unknown	IPF	Japanese RCT Capacity 004	Annual VC decline FVC change at 72 weeks	52 weeks 72 weeks	Reduced FVC decline
		IPF	ASCEND	Annual FVC change or death	52 weeks	Reduced FVC decline
		Non-IPF ILD (e.g., RA-ILD)	Multinational RCT	FVC change at 24 weeks	24 weeks	Reduced FVC decline
		RA-ILD	TRAIL1	Annual FVC decline >10% or death	52 weeks	Awaited
Rituximab	B cell depletion	RD-ILD	RECITAL	FVC change at 24 weeks	48 weeks	Awaited
Tocilizumab	IL-6 blockade	SSc-ILD	faSScinate	Modified Rodnan skin score; FVC decline as a secondary outcome	24 weeks	Numerically greater reduction of skin fibrosis; seemed to reduce FVC decline
		SSc-ILD	FocuSSced	Modified Rodnan skin score; FVC decline as a secondary outcome	48 weeks	No primary endpoint was met; seemed to reduce FVC decline

FVC = forced vital capacity, ILD = interstitial lung disease, IPF = idiopathic pulmonary fibrosis, RD = rheumatic disease, RA = rheumatoid arthritis, RCT = randomized controlled trial, SSc = systemic sclerosis.

**Table 2 pharmaceuticals-14-00251-t002:** Screening algorithm for SSc-PAH proposed in 2013 ^a^.

			Quality of Evidence
All patients with SSc should be screened for PAH			Moderate
Initial screening evaluation in patients with SSc or SSc-spectrum disorders			
► Pulmonary function test (PFT) with diffusion capacity of carbon monoxide (DLCO)			High
► Transthoracic echocardiography (TTE)			High
► N-terminal probrain natriuretic peptide (NT-proBNP)			Moderate
► DETECT algorithm if DLCO% < 60% and >3 years disease duration from non-RP			Moderate
Recommendations for right heart catheterization for SSc or SSc-spectrum disorders			
Screening method	Parameter cut-off	Signs/symptoms requirement ^b^	
PFT ^c^	FVC/DLCO ratio > 1.6 and/or DLCO < 60%	Yes	High
	FVC/DLCO ratio > 1.6 and/or DLCO < 60% plusNT-proBNP > 2 x upper limit of normal	No	High
TTE	TR velocity	Yes	High
	2.5–2.8 m/s	No	High
	>2.8 m/s	No	High
	Cavity enlargements irrespective of TR velocityRA major dimension > 53 mm orRV mid-cavity dimension > 35 mm		
Composite	Meets DETECT algorithm with DLCO% < 60% and >3 years of disease duration	No	Moderate

^a^ Cited and modified from “Recommendations for screening and detection of connective tissue disease-associated pulmonary arterial hypertension” by Khanna D, et al. Arthritis Rheum 2013; 65: 3194–3201 [99]. ^b^ Symptoms: dyspnea upon rest or exercise, fatigue, presyncope/syncope, chest pain, palpitations, dizziness, lightheadedness. Signs: loud pulmonic sound, peripheral edema. ^c^ Without overt systolic dysfunction, greater than grade I diastolic dysfunction, greater than mild mitral or aortic valve disease, or evidence of PAH in echocardiography. DLCO = diffusion capacity of carbon monoxide; FVC = forced vital capacity; NT-proBNP = N-terminal probrain natriuretic peptide; PAH = pulmonary arterial hypertension; PFT = pulmonary function test; RA = right atrium; RV = right ventricle; SSc = systemic sclerosis; TTE = transthoracic echocardiography.

**Table 3 pharmaceuticals-14-00251-t003:** Efficacy of drug monotherapy for group 1 PAH according to WHO functional class ^a^.

			Class ^b^-Level ^c^					
			WHO-FC II		WHO-FC III		WHO-FC IV	
ERA	Ambrisentan		I	A	I	A	IIb	C
	Bosentan		I	A	I	A	IIb	C
	Macitentan		I	B	I	B	IIb	C
PDE5i	Sildenafil		I	A	I	A	IIb	C
	Tadalafil		I	B	I	B	IIb	C
	Vardenafil		IIb	B	IIb	B	IIb	C
sGC stimulator	Riociguat		I	B	I	B	IIb	C
PCA	Epoprostenol	Intravenous	‒	‒	I	A	I	A
	Iloprost	Inhaled	‒	‒	I	B	IIb	C
		Intravenous	‒	‒	IIa	C	IIb	C
	Treprostinil	Subcutaneous	‒	‒	I	B	IIb	C
		Inhaled	‒	‒	I	B	IIb	C
		Intravenous	‒	‒	IIa	C	IIb	C
		Oral	‒	‒	IIb	B	‒	‒
	Beraprost		‒	‒	IIb	IIb	‒	‒
	Selexipag [oral]		I	B	I	B	‒	‒

^a^ Cited and modified from “2015 European Society of Cardiology (ESC)/European Respiratory Society (ERS) guidelines for the diagnosis and treatment of pulmonary hypertension” by Galiè N et al. Eur Heart J. 2016; 37:67–119 [97], ^b^ Class of recommendation, ^c^ level of evidence, ERA = endothelin receptor antagonist; PDE5i = phosphodiesterase 5 inhibitor; sGC = soluble guanylate cyclase; PAH = pulmonary arterial hypertension; PCA = prostacyclin analog or receptor antagonist; WHO = World Health Organization; FC = functional class.

**Table 4 pharmaceuticals-14-00251-t004:** 2015 European Society of Cardiology (ESC)/European Respiratory Society (ERS) guidelines for the risk assessment of patients with PAH ^a^.

Parameters of Prognosis	Estimated 1-Year Mortality		
	Low Risk < 5%	Low Risk < 5%	Low Risk < 5%
Clinical signs of right heart failure	Absent	Absent	Present
Progression of symptoms	No	Slow	Rapid
Syncope	No	Occasional syncope ^b^	Repeated syncope ^c^
WHO functional class	I, II	III	IV
6MWD	>440 m	165–440 m	<165 m
Cardiopulmonary exercise testing	Peak VO_2_ > 15 mL/min/kg (>65% predicted) VE/VCO_2_ slope < 36	Peak VO_2_ >15 mL/min/kg (35–65% predicted) VE/VCO_2_ slope < 36	Peak VO_2_ <11 mL/min/kg (<35% predicted) VE/VCO_2_ slope ≥ 45
NT-proBNP levels	BNP < 50 ng/L NT-proBNP < 300 ng/L	BNP 50–300 ng/L NT-proBNP 300–1400 ng/L	BNP >300 ng/L NT-proBNP > 1400 ng/L
Imaging (echocardiography, CMR imaging)	RA area <18 cm^2^ No pericardial effusion	VE/VCO_2_ slope < 36	RA area > 26 cm^2^ pericardial effusion
Hemodynamics	RAP < 8 mmHg CI ≥ 2.5 L/min/m^2^ S_v_O_2_ > 65%	RAP 8–14 mmHg CI 2.0–2.4 L/min/m^2^ S_v_O_2_ 60–65%	RAP > 14 mmHg CI < 2.0 L/min/m^2^ S_v_O_2_ < 60%

^a^ Cited and modified from “2015 European Society of Cardiology (ESC)/European Respiratory Society (ERS) guidelines for the diagnosis and treatment of pulmonary hypertension” Galiè N, et al. Eur Heart J. 2016;37:67–119 [97]. ^b^ Occasional syncope during brisk or heavy exercise, or occasional orthostatic syncope in an otherwise stable patient. ^c^ Repeated episodes of syncope even with little or regular physical activity. 6MWD = 6-min walking distance; BNP = brain natriuretic peptide; CI = cardiac index; CMR = cardiac magnetic resonance; NT-proBNP = N-terminal probrain natriuretic peptide; PAH = pulmonary arterial hypertension; RA = right atrium; RAP = right atrial pressure; S_v_O_2_ = mixed venous oxygen saturation; VE/VCO_2_ = ventilatory equivalents for carbon dioxide; VO_2_ = oxygen consumption; WHO = World Health Organization.

## Data Availability

Not applicable.

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
