# Peer review of "Pharmacological Interventions for Pulmonary Involvement in Rheumatic Diseases"

_pharmaceuticals, 2021, doi:10.3390/ph14030251_

Round 1
Reviewer 1 Report
Nice review
Author Response
We thank the reviewer.
Reviewer 2 Report
In this narrative review on pharmacological interventions for lung involvement in RMD, the Authors aimed to summarize the current approach to treatment of both ILD and PAH in patients with systemic autoimmune diseases. Although this is an important topic, several similar reviews are being published every month. This review does not add significant contribution to the field in the present form. As such, I think it should be improved.
Some major points:
- The Authors focused on SSc, myositis and RA associated pulmonary involvement, but other diseases in which lung involvement is prominent, such as MCTD and Sjogren syndrome, are not being reviewed here.
- As the Authors wrote in the abstract, this review aims to provide insight on the updated treatment strategies adopting targeted agents in RMD associated ILD. As such, I would expect a more linear discussion regarding the different strategies that have been proposed (which drug in first line, for example, and why) as well as an overview on the ongoing debate of such strategies (see for example 10.1016/S2665-9913(19)30144-4).
- Finally, I suggest to add a paragraph on possible future treatments, based on preliminary research and/or ongoing trials, as well as some important pathophysiological and pharmacodynamic considerations of targeted therapies in RMD-ILD.
Some minor but still important points:
- Not all ILDs associated to RMDs are progressive. The Authors should mention which definition of progressive ILD has been used, for example, in the inclusion criteria of the nintedanib trials, and should acknowledge that different definitions of progressive ILD have been proposed (see 10.1183/13993003.02026-2019)
- Some recent trials have been missed, such as the FocuSSced phase III trial for TCZ (10.1016/S2213-2600(20)30318-0); the Authors should also mention that in the TCZ trials lung function was not the primary endpoint but a secondary endpoint in an overall negative trial.
- Page 5 line 199: Nintedanib is now the first approved treatment for SSc-ILD (both FDA and EMA). As such, I would give more space to nintedanib and its benefits in SSc-ILD, also considered that several post hoc analyses of the SENSCIS (misspelled in the text) trial are now available (see 10.1002/art.41576 and 10.1016/S2213-2600(20)30330-1)
- Page 5 line 202: the two ongoing trials of pirfenidone for SSc-ILD (NCT03856853 and NCT03221257) should be mentioned and its rationale discussed.
- Table 1. The column "proven efficacy" could be more informative to the reader, for example by adding the timepoint of outcome assessment (i.e. 52 weeks, etc) and specifying if the lung outcome was a primary or a secondary endpoint, etc
- Page 8-9, PAH treatment. Selexipag should be discussed in the text.
English is fine but minor spell checks are needed.
Author Response
Please see the attachment
Some major points:
- The Authors focused on SSc, myositis and RA associated pulmonary involvement, but other diseases in which lung involvement is prominent, such as MCTD and Sjogren syndrome, are not being reviewed here.
[Response]
We agree with the reviewer that the lung involvement, particularly ILD, is prominent in MCTD and Sjogren’s syndrome. However, their data on pulmonary treatment are far more limited compared to the diseases chosen for our review, despite the growing epidemiologic knowledge on their pulmonary manifestations. In this context, we added a separate section to describe pulmonary manifestations of MCTD and Sjogren’s syndrome as follows.
Lines 545-587
- Pulmonary manifestations of other rheumatic diseases
Other than RA, SSc, and myositis, Sjogren’s syndrome (SjS) and mixed connective tissue disease (MCTD) also show a substantial prevalence of pulmonary manifestations including ILD and PAH. Although data on the epidemiology are growing in SjS and MCTD, RCT data regarding treatments are very limited. However, we may be able to apply similar treatment strategies in SjS and MCTD as in other RDs described in this review, based on the dominant underlying mechanism of fibrosis versus inflammation or chronic progression versus acute exacerbation.
3.1. Sjogren’s syndrome associated pulmonary involvements
SjS is a systemic autoimmune disease primarily affecting exocrine glands, which ultimately leads to the destruction of the given tissue [113]. The autoimmunity of SjS activates both immune cells and glandular epithelial cells showing a histological lesion of focal lymphocytic infiltrates, enriched in CD4+ T-cells, around the salivary and lachrymal ducts and even in lung tissues [114]. Of note, the focus score of salivary glands has been shown to correlate with increased prevalence of airway disease and ILD in SjS [115]. These findings may suggest that the glandular and extra-glandular lesions of SjS share similar pathogenic pathways involving autoimmunity to epithelial cells.
The prevalence of pulmonary involvement in patients with SjS has been estimated to range from 10% to 20% [116]. Airway disease and ILD are predominant forms of lung involvement among the pulmonary manifestations of SjS, but lymphoproliferative disorders as well as cystic lesions are also observed [116]. The most common histologic type of ILD associated with SjS was found to be NSIP followed by UIP and LIP [117,118], with NSIP present in up to 45% of biopsied cases [117] and mostly of fibrotic type [118]. Unlike RA-UIP, SjS-UIP tends to have a better response to immunotherapy as compared to IPF [119]. LIP constitutes 15% of SjS-ILD, of which clinical course is variable from complete resolution without treatment to progression and possible death or transformation to lymphoma [120].
3.2. Mixed connective tissue disease associated pulmonary involvements
MCTD is characterized by a mixed feature of two or more RDs including but not limited to SSc, myositis or systemic lupus erythematosus, with disease specific high titer anti-U1 RNP antibodies [121]. The common pulmonary manifestations of MCTD are ILD and PAH. A Norwegian nation-wide cross- sectional study showed that 35% of MCTD patients had lung fibrosis in HRCT after a mean disease duration 9 years [122]. A higher rate of ILD was found (up to two thirds of the patients) in the hospital-based study [123]. In the Norwegian study, 19% of MCTD associated ILD (MCTD-ILD) was severe in extent involving > 50% of lung parenchyma [122]. However, the retrospective cohort study on unselected 53 patients with a mean disease duration of 9 years in Brazil showed that 51% of the patients had ILD at baseline with a mean FVC that remained stable at around 77% over 10 years of follow-up and a mean DLCO that declined from 84% to 71% [124]. Such discrepant findings leave a debate on the natural course of MCTD-ILD, and further cohort studies at a larger scale are in need. Together with ILD, PAH is a major prognostic factor of MCTD [125, 126]. The Norwegian population-based prevalence of pulmonary hypertension was reported to be less than 5% over 5.6 years [125], but PAH has been recognized as the leading cause of death in patients with MCTD, explaining 41% of all deaths [126].
- As the Authors wrote in the abstract, this review aims to provide insight on the updated treatment strategies adopting targeted agents in RMD associated ILD. As such, I would expect a more linear discussion regarding the different strategies that have been proposed (which drug in first line, for example, and why) as well as an overview on the ongoing debate of such strategies (see for example 10.1016/S2665-9913(19)30144-4).
[Response]
We thank the reviewer for the suggestion to present more hierarchical recommendations together with the rationale behind such recommendations for RD-ILD. We strongly agree to the importance of such approach, and revised the RA-ILD and SSc-ILD parts to meet this approach as follows.
Line 118-226
- Pharmacologic treatment of RA-ILD
Unfortunately, there have been no randomized controlled trials (RCTs) for RA-ILD and the EULAR and ACR recommendations do not specify how to treat RA-ILD yet [16, 17]. In general, physicians either adopt the treatment strategies against the corresponding pattern of idiopathic interstitial pneumonia or practice empirical therapies with immunosuppressives. However, due to the heterogeneous clinical behaviors of RA-ILD, the treatment goal is hard to define: treat to achieve recovery versus to stabilize or slow progression. In case of acute exacerbation, most therapies are immunosuppressants with potent anti-inflammatory effects assuming reversibility of the lesions while in case of chronic progression, anti-fibrotic approach will take the priority [2]. Therefore, physicians need to understand the dominant pathogenic mechanism underlying the clinical behavior of individual RA-ILD, which can vary at different time points even in a same patient.
- Treatment with conventional agents
- Treatment with targeted agents
-
-
- General treatment strategy for RA-ILD
Due to the heterogeneous progression patterns across individuals, it is hard to define an optimal treatment strategy in RA-ILD. Although the baseline extent of lung injury has been acknowledged as the most reliable risk factor of both progression and acute exacerbation, the cut-off extent to initiate treatment is not well defined. The international guidelines on RA management have yet to specify when to initiate treatment and what to be the 1st line agent [16, 17]. Moreover, the treatment of acute exacerbation should be different from that of chronic progression [2]. However, it seems reasonable to use nintedanib when FVC loss is progressive enough to deteriorate symptoms (e.g. dyspnea or exercise capacity) or of when the current status is severe enough to qualify for the previous RCTs including the INBUILD trial (FVC of ≥45%, DLCO of 30-79%) [29].
Lines 257-367
- Pharmacologic treatment of SSc-ILD
The lung function of SSc patients with FVC of ≥ 80% at baseline rarely declines [43]. Thus, symptomatic patients are the primary target of treatments, particularly those whose ILD is moderate-to-severe or shows progression. The recent European consensus statements presented an agreement on screening of ILD for all SSc patients at baseline particularly in the presence of risk factors (diffuse cutaneous subset, anti-topoisomerase antibody, low DLCO) preferably by HRCT but also with pulmonary function test (PFT) and clinical assessment as supporting tools [48]. The statements recommend treatment for all severe cases defined by PFT or HRCT, and for those who progress based on clinical assessment, HRCT, and/or PFT. However, the statements did not specify the threshold to define severe ILD or when to initiate or escalate treatment. Those who have symptoms that are progressively deteriorating (newly developed functional class II and more), whose FVC and/or DLCO decline is large enough (≥ 10% and ≥ 15%, respectively) [49, 50], or whose FVC or DLCO is severe enough to justify treatment-related adverse events (refer to the inclusion criteria for Scleroderma Lung Study or SENSCIS trial below) [51-53] would be reasonable candidates to initiate or escalate treatment.
- Treatment with conventional agents
- Treatment with targeted agents
- Antifibrotics
- Biologics
1.2.2.3. General treatment strategy for SSc-ILD
For those who are asymptomatic with minimal extent of ILD (e.g., FVC>80%), watchful monitoring without treatment is justified [42, 43]. For those who need treatment, the EULAR guidelines endorse CYC over MMF as the 1st line treatment for SSc-ILD [55] while the European consensus statements consider CYC and MMF, and nintedanib as equivalent options for treatment initiation or escalation [48]. The subgroup analysis of SENSCIS showed a comparable FVC change between placebo with baseline MMF group (-66.5 mL/year) versus nintedanib without baseline MMF group (-63.5 mL/year) [53]. Thus, whether to use anti-fibrotics as a 1st line treatment needs further data (e.g., head-to-head comparison between immunosuppressants versus anti-fibrotics) and discussion. A reasonable approach would be to switch to or add nintedanib when refractory to CYC or MMF. Although some evidence emerged suggestive of the benefits with TCZ and RTX on lung function, such evidence is only exploratory and needs to be confirmed in an RCT that examines lung function as the primary end point.
- Finally, I suggest to add a paragraph on possible future treatments, based on preliminary research and/or ongoing trials, as well as some important pathophysiological and pharmacodynamic considerations of targeted therapies in RMD-ILD.
[Response]
We added the following section in the treatment section.
Lines 408-428
- Future treatment strategies for rheumatic disease associated interstitial lung disease
Nintedanib and pirfenidone slow but does not halt the progression of IPF. Thus, interests on combination of the two drugs are growing, but more data are needed on the safety and efficacy of combination therapy. There are a few studies that showed relatively acceptable tolerability and safety of the two drugs combined for a short-term period (12~24weeks) [37, 38]: combination was completed in two thirds of patients despite a higher adverse event rates compared to the single drug treatment. However, there has yet to be large randomized controlled trials assessing whether combination therapy has increased efficacy compared to treatment with a single antifibrotic drug.
Several clinical trials evaluating the efficacy of the new drugs on pulmonary outcomes are underway. ISABELA 1 and 2 trials are now ongoing to examine the effect of autotaxin inhibitor GB-0998 among patients with IPF (ClinicalTrials.gov number, NCT03711162; NCT03733444) [81]. A phase 2 RCT of bortezomib plus MMF versus MMF alone on patients with SSc-ILD is currently ongoing (ClinicalTrials.gov number, NCT02370693). Romilkimab, a monoclonal antibody targeting IL-4 and IL-13, showed efficacy after 24 weeks on skin fibrosis associated with diffuse SSc under background immunosuppressive agents in a phase 2 pilot study [82]. However, it failed to show benefits in the of IPF after 52 weeks [83]. Guselkumab (monoclonal antibody that blocks IL-23) is under investigation in SSc patients with skin fibrosis as the primary outcome of and lung function as one of the secondary outcomes (ClinicalTrials.gov number, NCT04683029).
Some minor but still important points:
- Not all ILDs associated to RMDs are progressive. The Authors should mention which definition of progressive ILD has been used, for example, in the inclusion criteria of the nintedanib trials, and should acknowledge that different definitions of progressive ILD have been proposed (see 10.1183/13993003.02026-2019)
[Response]
As suggested by the reviewer, we have added the definition of progression as well as the inclusion criteria of the key clinical trials.
Lines 155-157, 172, 184-185
- Pharmacologic treatment of RA-ILD
- Treatment with targeted agents
----------------------------------------------------------------------In the phase II proof-of-concept TOMORROW study on patients with IPF diagnosed based on biopsy and/or HRCT whose % predicted FVC of ≥ 50% and DLCO of 30-79% [25], -------------------------------------------------------------------------------------------------------------------------------------------------------------------------------------------------------------------------
The recent INBUILD study, an RCT that assessed the efficacy and safety of nintedanib in patients with non-IPF ILD (entry criteria: % predicted FVC of ≥ 45% and DLCO of 30-79%), included a subgroup of patients with RD-ILDs, mostly RA-ILD [29]. ------------------------------------------------------------------------------------------------------------------------------------------In the rest two CAPACITY trials (004 and 006) on patients with multinational backgrounds (entry criteria: % predicted FVC of 50%-90% and DLCO of 35-90%), --------------------------------------------------------------------------------------------------------------------------------------------------------------------------------------------
Lines 267-272, 279, 292-294
- Pharmacologic treatment of SSc-ILD
The lung function of SSc patients with FVC of ≥ 80% at baseline rarely declines [43]. Thus, symptomatic patients are the primary target of treatments, particularly those whose ILD is moderate-to-severe or shows progression. ---------------------------------------------------------------------------------------------------------------------------------------------------------------------- Those who have symptoms that are progressively deteriorating (newly developed functional class II and more), whose FVC and/or DLCO decline is large enough (≥ 10% and ≥ 15%, respectively) [49, 50], or whose FVC or DLCO is severe enough to justify treatment-related adverse events (refer to the inclusion criteria for Scleroderma Lung Study or SENSCIS trial below) [51-53] would be reasonable candidates to initiate or escalate treatment.
- Treatment with conventional agents
In the Scleroderma Lung Study (SLS) I on 158 SSc patients with symptomatic ILD (active alveolitis on imaging and FVC ranging between 45 –85%) [51],-----------------------------------------------------------------------------------------------------------------------------------------------------------------------------------------------
- Treatment with targeted agents
- Antifibrotics
The SENSCIS trial was performed on 576 patients with SSc-ILD (mean age 54.6 years, 74-76% female) of at least 10% extent of the lung [53]. The entry criteria of ILD required % predicted FVC ≥ 40% and % predicted DLCO ranging in 30-90%.
- Some recent trials have been missed, such as the FocuSSced phase III trial for TCZ (10.1016/S2213-2600(20)30318-0);the Authors should also mention that in the TCZ trials lung function was not the primary endpoint but a secondary endpoint in an overall negative trial.
[Response]
We have updated the corresponding trial with its outcome definitions.
Lines 323-325, 329-341
In the phase II faSScinate trial that assessed the skin fibrosis as the primary outcome and lung function as the secondary outcome, 48-week treatment with tocilizumab (TCZ) provided an encouraging numerical improvement in the modified Rodnan skin scores and evidence of less decline in lung function [59].--------------------------------------------------------------------------------------------------------------------------------------------------------------------------------------- In the subsequent focuSSced study of the multinational background on 210 patients with early diffuse cutaneous SSc (duration ≤ 5 years) [61], TCZ and placebo were compared regarding skin fibrosis (primary outcome) and lung function (secondary outcome). After 48 weeks, the skin fibrosis endpoint was not met but the FVC loss was in favor of TCZ group than placebo (between group difference, 241 mL, p=0.002). The results of focuSSced study on FVC should be interpreted with caution because the SSc-ILD was not the primary outcome and only 65% of enrolled patients had SSc-ILD at baseline.
- Page 5 line 199: Nintedanib is now the first approved treatment for SSc-ILD (both FDA and EMA). As such, I would give more space to nintedanib and its benefits in SSc-ILD, also considered that several post hoc analyses of the SENSCIS (misspelled in the text) trial are now available (see 10.1002/art.41576 and 10.1016/S2213-2600(20)30330-1)
[Response]
We have added a paragraph for SENSCIS and its post-hoc and long-term analyses.
Lines 290-313
1.2.2.2. Treatment with targeted agents
- Antifibrotics
The SENSCIS trial was performed on 576 patients with SSc-ILD (mean age 54.6 years, 74-76% female) of at least 10% extent of the lung [53]. The entry criteria of ILD required % predicted FVC ≥ 40% and % predicted DLCO ranging in 30-90%. The % predicted FVC of included patients was 72% and DLCO 53% at baseline. 48.4% of the study patients were taking background MMF. After 52 weeks of treatment, nintedanib (150mg twice daily) was associated with a lower annual rate of FVC decline compared to placebo treatment (-52.4 mL/year versus -93.3 mL/year). The curves for the FVC change from the baseline separated by 12 weeks and continued to diverge until the end of study. The effect of nintedanib was additive when combined with MMF: the annual rates of change in FVC among the patients receiving mycophenolate were −40.2 mL/year in the nintedanib group and −66.5 mL/year in the placebo group, and the corresponding rates among the patients who were not receiving mycophenolate were −63.9 mL/year and −119.3 mL/year. Of note, the SENSCIS trial did not show any treatment effect on the skin fibrosis. According to the subgroup analysis, the effect of nintedanib was consistent regardless of the use of MMF, suggesting that there is no synergy but additive effect between nintedanib and MMF [56]. The gastrointestinal adverse (e.g., diarrhea) events were more common in the SENSCIS trial compared to the INPULSIS trials probably due to the gastrointestinal involvement by SSc itself [26, 53]. The post-hoc analysis on the SENSCIS trial showed that the effect of nintedanib was significant to prevent % predicted FVC decline of >10% but not of >5% [57]. An open label extension study is ongoing to provide long term data on nintedanib use in patients with SSc-ILD (ClinicalTrials.gov number, NCT03313180). Based on the SENSCIS trial [53], nintedanib was approved to treat SSc-ILD by FDA in 2019 and by European Medicines Agency (EMA) in 2020.
- Page 5 line 202: the two ongoing trials of pirfenidone for SSc-ILD (NCT03856853 and NCT03221257) should be mentioned and its rationale discussed.
[Response]
We have introduced the ongoing trials of pirfenidone. Since the additive effect of nintedanib and MMF was described in the previous paragraph, the description of the rationale part was not replicated.
Lines 314-321
Acceptable safety and tolerability of the anti-fibrotic agent pirfenidone were reported among patients with SSc-ILD [58], but its efficacy needs to be assessed in further clinical trials. There are two ongoing RCTs on SSc-ILD. The efficacy and safety of pirfenidone is now being assessed in a phase 3 study on 144 SSc-ILD patients (ClinicalTrials.gov number, NCT03856853). In addition, the SLS III study (phase 2) will randomize 150 patients with SSc-ILD to pirfenidone versus placebo, with background MMF, over 18 months of follow-up (ClinicalTrials.gov number, NCT03221257), to assess the effect of combination treatment of pirfenidone and MMF versus MMF alone.
- Table 1. The column "proven efficacy" could be more informative to the reader, for example by adding the timepoint of outcome assessment (i.e. 52 weeks, etc) and specifying if the lung outcome was a primary or a secondary endpoint, etc
[Response]
The Table 1 was revised as follows.
Table 1. Recent clinical trials on targeted drugs for idiopathic pulmonary fibrosis and rheumatic disease-associated interstitial lung disease
|
Targeted therapy |
Mechanism of therapy |
Target disease |
Key RCT name |
Primary outcome |
Treatment duration |
Proven efficacy |
|
Nintedanib |
Tyrosine kinase inhibition |
IPF |
TOMORROW[25] |
Annual FVC decline |
52 weeks |
Reduced FVC decline |
|
|
|
IPF |
INPULSIS 1 & 2[26] |
Annual FVC decline |
52 weeks |
Reduced FVC decline |
|
|
|
Non-IPF ILD (e.g., RA-ILD) |
INBUILD[29] |
Annual FVC decline |
52 weeks |
Reduced FVC decline |
|
|
|
SSc-ILD |
SENSCIS[50] |
Annual FVC decline |
52 weeks |
Reduced FVC decline |
|
Pirfenidone |
Unknown |
IPF |
Japanese RCT[30] Capacity 004 [31] |
Annual VC decline FVC change at 72 weeks |
52 weeks 72 weeks |
Reduced FVC decline |
|
|
|
IPF |
ASCEND[32] |
Annual FVC change or death |
52 weeks |
Reduced FVC decline |
|
|
|
Non-IPF ILD (e.g., RA-ILD) |
Multinational RCT[33] |
FVC change at 24 weeks |
24 weeks |
Reduced FVC decline |
|
|
|
RA-ILD |
TRAIL1[34] |
Annual FVC decline >10% or death |
52 weeks |
Awaited |
|
Rituximab |
B cell depletion |
RD-ILD |
RECITAL[65] |
FVC change at 24 weeks |
48 weeks |
Awaited |
|
Tocilizumab |
IL-6 blockade |
SSc-ILD |
faSScinate[59] |
Modified Rodnan skin score; FVC decline as a secondary outcome |
24 weeks |
Numerically greater reduction of skin fibrosis; seemed to reduce FVC decline |
|
|
|
SSc-ILD |
FocuSSced[61] |
Modified Rodnan skin score; FVC decline as a secondary outcome |
48 weeks |
No primary endpoint was met; seemed to reduce FVC decline |
- Page 8-9, PAH treatment. Selexipag should be discussed in the text.
[Response]
We have added the following text on selexipag.
Lines 513-518
2.1.2. Pharmacological treatment of SSc-PAH
2.1.2.2. Treatment with targeted agents and risk stratification
-----------------------------------------------------------------------------------------------------------------------------------------------------------------------------------Although not commented in EULAR recommendations, selexipag, an oral selective prostacyclin IP receptor agonist, has shown efficacy in the phase 3 GRIPHON trial of reducing the risk of a primary composite endpoint of morbidity and mortality by 40% and 41% in PAH and RD-PAH (majority, SSc-PAH), respectively [106, 107]. Selexipag is the only oral prostacyclin receptor agonist that showed a reduction of morbidity and mortality in a RCT.
- English is fine but minor spell checks are needed.
[Response]
We have spell-checked.
Reviewer 3 Report
This review discuss the pharmacological interventions for pulmonary involvement in rheumatic diseases.
Comments:
Pulmonary involvement in various rheumatic diseases should be described in more detail.
The authors should discuss pharmacokinetic parameters as well as the adverse effects of drugs used in the therapy of pulmonary involvement in rheumatic diseases.
Abbreviations should be explained.
Author Response
This review discuss the pharmacological interventions for pulmonary involvement in rheumatic diseases.
- Pulmonary involvement in various rheumatic diseases should be described in more detail.
[Response]
Please refer to our answer to the 1st comment of the reviewer #1 that additionally describes pulmonary manifestations of Sjogren’s syndrome and mixed connective tissue disease. We also have added the following paragraphs.
Lines 85-88, 228-230
- Rheumatoid arthritis associated ILD (RA-ILD)
Potential targets of lung injury in RA include almost all components of lung structures. Thus, lung injury associated with RA encompass a wide spectrum of disorders such as parenchymal (ILD), airway (bronchiectasis or bronchiolitis), pleural (pleurisy), and vascular diseases. Among them, ILD is most common.
- Systemic sclerosis associated interstitial lung disease (SSc-ILD)
The pulmonary manifestations of SSc can be both direct and indirect. The former category includes parenchymal (ILD) and vascular diseases (PAH) and the latter includes aspiration due to gastroesophageal reflux associated with esophageal sphincter fibrosis.
- The authors should discuss pharmacokinetic parameters as well as the adverse effects of drugs used in the therapy of pulmonary involvement in rheumatic diseases.
[Response]
We have added the following section.
Lines 197-215
- Pharmacologic treatment of RA-ILD
- Treatment with targeted agents
----------------------------------------------------------------------------------------------------------------------
There is an overlap of adverse event profiles between nintedanib and pirfenidone [35]. In patients treated with nintedanib, gastrointestinal adverse events (e.g. diarrhea) were most common with mild-to-moderate intensity, accounting for the majority of the drug discontinuation [25, 26, 29]. Similarly, the most common adverse event of pirfenidone was gastrointestinal including nausea and vomiting, experienced by 40% and 18% of treated patients, respectively, out of 2,059 person-years of exposure [36]. However, the two drugs have different pharmacokinetic profiles. Nintedanib is metabolized predominantly by ester cleavage and then glucuronidated to be excreted via the biliary system [35]. Use of nintedanib is associated with liver function test abnormalities in less than 5% of patients, and is not recommended for those with moderate-to-severe hepatic dysfunction. Regular liver function monitoring is required. Nintedanib has a low potential for drug-drug interactions, especially with drugs metabolized by cytochrome P450 enzymes. Pirfenidone is metabolized by various cytochrome P450 enzymes in the liver and predominantly excreted via the urine [35]. Similar rates of liver function test abnormalities were observed with pirfenidone as with nintedanib [25, 26, 31, 32]. Pharmacokinetically, no drug-drug interaction was observed between nintedanib and pirfenidone. There are RCTs that examined the effect of nintedanib added on pirfenidone treatment [37, 38]. More reports of nausea and vomiting were observed with nintedanib added on pirfenidone than used alone. But most of them were mild-to-moderate as with the single drug treatment and the combination did not provide a new safety signal.
- Abbreviations should be explained
[Response]
All of the abbreviations were explained in the text, and a list of abbreviation was added before the references.
Round 2
Reviewer 2 Report
I have no further comments.
Reviewer 3 Report
The revised version of the manuscript is suitable for publication.